# Novel Bacteriophages Show Activity against Selected Australian Clinical Strains of *Pseudomonas aeruginosa*

**DOI:** 10.3390/microorganisms10020210

**Published:** 2022-01-19

**Authors:** Samuel Namonyo, Gilda Carvalho, Jianhua Guo, Karen D. Weynberg

**Affiliations:** 1Australian Centre for Water and Environmental Biotechnology (ACWEB, formerly AWMC), The University of Queensland, St. Lucia, QLD 4072, Australia; s.namonyo@uq.net.au (S.N.); j.guo1@uq.edu.au (J.G.); 2Queensland Alliance for Environmental Health Services, The University of Queensland, Woolloongabba, QLD 4102, Australia; 3Australian Centre for Ecogenomics, School of Chemistry & Molecular Biosciences, The University of Queensland, St. Lucia, QLD 4072, Australia

**Keywords:** bacteriophage, *Pseudomonas aeruginosa*, phage therapy

## Abstract

Multi-drug resistant (MDR) clinical strains of *Pseudomonas aeruginosa* are the most prevalent bacteria in the lungs of patients with cystic fibrosis (CF) and burn wounds and among the most common in immunocompromised hospital patients in Australia. There are currently no promising antibiotics in the pipeline being developed against these strains. Phage therapy, which uses viruses known as bacteriophages to infect and kill pathogenic bacteria, could be a possible alternative treatment. To this end, we isolated and characterised four novel phages against Australian clinical strains of *P. aeruginosa* isolated from patients with cystic fibrosis, from infected blood and joint aspirate in Southeast Queensland, Australia. Activated sludge was enriched for phages using the clinical strains, and four bacteriophages were isolated. The phages were able to cause lysis in a further three identified clinical isolates. Morphology showed that they were all tailed phages (of the order Caudovirales), two belonging to the family Myoviridae and the others assigned to the Podoviridae and Siphoviridae. Their genomes were sequenced to reveal a doubled stranded DNA topology with genome sizes ranging from 42 kb to 65 kb. In isolating and characterising these novel phages, we directed our efforts toward the development and use of these phages as candidates for phage therapy as an alternative strategy for the management or elimination of these pathogenic strains. Here we describe novel phage candidates for potential therapeutic treatment of MDR Australian clinical isolates of *P. aeruginosa*.

## 1. Introduction

Antimicrobial resistance (AMR) to conventional antibacterial compounds (e.g., β-lactams, rifamycins, aminoglycosides etc.) has soared, especially in treating hospital-acquired (nosocomial) infections [1,2]. The ‘ESKAPE’ pathogens, identified by the Infectious Diseases Society of America [2] as the most critical AMR species, consist of *Enterococcus faecium*, *Staphylococcus aureus*, *Klebsiella pneumoniae*, *Acinetobacter baumannii*, *Pseudomonas aeruginosa* and *Enterobacter* spp. and represent a major global health threat [3]. In 2017, the World Health Organisation listed five of these in the priority list for which new antibiotics are urgently needed. Without alternative treatments these AMR pathogens are projected to present an economic cost of 100 billion dollars and approximately 10 million deaths every year by 2050 [4,5,6]. Of the above ESKAPE pathogens, *P. aeruginosa* is the most highly polytropic, which is the reason for their ecological, medical, agricultural and commercial relevance [7]. *P. aeruginosa* has an array of adaptable features and mechanisms to survive, persist and resist a wide range of antibacterial therapies. These include, but are not limited to, exotoxin A, flagella, pili, proteases, phospholipases, lectins, siderophores, pyocyanin, lipopolysaccharides and formation of biofilms [8,9]. *P. aeruginosa* is increasingly receiving attention as a nosocomial pathogen because of its ubiquitous nature, intransigence and ability to infect almost any organ of an immunocompromised or immunodeficient individual [9]. *P. aeruginosa* has emerged as one of the principal causes of morbidity and mortality among health care associated (nosocomial) infections [8], most notably as the most common pathogen in the lungs of patients suffering from CF and those with burn wound infections [9,10,11].

With increasing cases of resistance of *P. aeruginosa* to antibiotics [12], methods such as phage therapy are being increasingly explored as promising alternatives [13]. Phages against *P. aeruginosa* were first isolated in the mid-twentieth century [14] with a major discovery of the Lindberg’s set of phages for serotyping purposes in the early to mid-1970s [15]. Phage therapy against *P. aeruginosa* has been done with remarkable success both in vivo and in vitro. Clinicians in the Eastern European block have successfully applied phage therapy against *P. aeruginosa* [16,17]. However, this has not been replicated in the West, due to factors including the widespread use of antibiotics and the lack of clinical guidelines from the regulatory bodies involved, due to ethical and safety concerns regarding the development of phage resistance by the bacteria in patients [18]. This has meant that phage therapy is mostly used as a last resort when all antibiotics have failed [19]. Most of the successful studies that have been conducted tended to utilize phage cocktails rather than a single phage, with the most successful therapies being those that combined phage cocktails (where two or more phages are used) in combination with antibacterial chemicals [13]. These therapies have had both animal and human trials, as reviewed by Rossito et al. [20].

As of 2015, Pires et al. [13] noted that there were 137 whole *Pseudomonas* bacteriophage genomes in public databases, of which about 95% were members of the tailed Caudovirales order and 85% of the tailed phages were specific for *P. aeruginosa* species. Of the 137, 4 belonged to the Cystoviridae, 2 to the Inoviridae and 2 to the Leviviridae, all these 3 having no tails. The case for phage therapy was advanced when in 2009, Merabishvili et al. [21] published a quality-controlled clinical trial with a phage cocktail (BFC-1) that could be used in clinical trials against burn wounds infected with *P. aeruginosa*. The cocktail formulations were well characterised in this case, and no adverse side effects were reported [21,22]. Sacher et al. [23] outlined the compassionate use of phage therapy (cPT) in Australia with phage products such as AB-PA01 [24] being applied against clinical isolates of MDR *P. aeruginosa*. Khawaldeh et al. [25] applied phages against an Australian MDR isolate of *P. aeruginosa* in a patient with refractory urinary tract infection. In Australia, the Therapeutic Goods Administration (TGA), a major public health institute, regulates the use of compassionate phage therapy recognising its potential in case of antibiotic resistance or failure [23,26]. However, there is a need to improve the bank of available phages, as well as the knowledge about their performance, towards a more widespread application of phage therapy.

The purpose of this study was to isolate and characterise novel phages against Australian MDR clinical isolates of *P. aeruginosa*, which could potentially be used in phage therapy as an alternative to antibiotics. We used clinical isolates from the Cystic Fibrosis Services Queensland Children’s Hospital (CFS-QCH), Brisbane, Australia, to isolate phages from activated sludge. Previous reports have identified geographic specificity of *P. aeruginosa* strains [27,28], and we hypothesised the novel phage isolates in this study may therefore be more suited to treat Australian clinical strains. The isolated phages were characterised to ascertain if they had properties desirable for use in phage therapy. Pursuant to their physical characterisation, their DNA was extracted, sequenced and screened for host genetic elements among other undesirable genes.

## 2. Materials and Methods

### 2.1. Bacterial Host Strains and Culture Conditions

Pure cultures of clinical strains of MDR *P. aeruginosa* (Table 1) isolated from patients from Southeast Queensland were sourced from the biobank at the Cystic Fibrosis Services Queensland Children’s Hospital (CFS-QCH), Brisbane, Australia.

The bacterial strains were cultured on Luria–Bertani (LB) broth medium at 37 °C [29]. Short term storage of the cultures was at 4 °C, and long-term storage was on 50% glycerol at −80 °C.

### 2.2. Isolation and Enrichment of the Bacteriophages

Activated sludge samples were collected from the Luggage Point wastewater treatment facility (Brisbane, QLD, Australia) in sterile capped 50 mL Falcon tubes (Corning Science, Reynosa, Mexico) and transported in cooler boxes to the laboratory. The samples were allowed to settle at 4 °C to separate large solids. A modified protocol of Van Twest et al. [30] was used to process the samples. Briefly, TBS tryptic soy broth (BD Difco, Franklin Lakes, NJ, USA) solution (with 10 mM CaCl_2_) was added to the supernatant at a ratio of 1:10 and incubated at room temperature with sequential mixing by gentle inversion for an hour. Samples were then centrifuged for 10 min at 10,000× *g* and filtered through a 0.22 μm membrane filter (Merck Millipore, Burlington, MA, USA), and the filtrate was recovered. A spot test, as per Van Twest et al. [30], was performed with the enriched phage filtrate and a control with sterile LB broth (BD Difco, Franklin Lakes, NJ, USA) to ascertain that the phages were present and active against clinical strains of *P. aeruginosa* mentioned above.

### 2.3. Purification and Enumeration of the Bacteriophages

A plaque assay was performed using the double agar overlay plaque assay [31] three times to purify the isolated phages. Briefly, 0.1 mL of the enriched phage filtrate was added to 0.3 mL of mid exponential *P. aeruginosa* AUS34 culture and 3 mL of low melting point top agar (Invitrogen, Waltham, MA, USA) and poured on top of a plate with solidified LB agar plate (BD Difco, Franklin Lakes, NJ, USA). This was then inverted and incubated for 24 h at 37 °C. One colony was picked aseptically with a pipette tip and suspended in 1 mL SM buffer (MP Biomedicals, Seven Hills, NSW, Australia) (NaCl 100 mM, MgSO_4_·7H_2_O 8 mM, Tris-Cl 50 mM) and mixed well by vortexing. This was then filtered through the 0.22 μm membrane filter. This process was performed twice more to obtain a pure phage stock. The pure phage stock was further enriched by incubation in liquid LB broth (BD Difco, Franklin Lakes, NJ, USA) and filtered through a 0.22 μm membrane filter (Merck Millipore, Burlington, MA, USA). A serial dilution was finally performed on the phage stock to determine the phage titre in PFU/mL calculated by Number of PFUs ÷ 1/dilution factor [32]. This was applied for the other three isolated phages using the three other bacterial strains and the isolated phages designated as Pseudomonas phage_AUS034, Pseudomonas phage_AUS260, Pseudomonas phage_AUS301 and Pseudomonas phage_AUS391.

### 2.4. Phage Morphology by Negative Staining and TEM

The four phages were further concentrated by ultrafiltration through a 100k MWCO Vivaspin 500 membrane filter (Sartorius, Göttingen, Germany) according to the manufacturer’s instructions. Then, 4 μL of the enriched phage filtrate was stained with 2% uranyl acetate (*w*/*v*) on a carbon coated copper grid (ProSciTech, Kirwan, QLD, Australia). Finally, these were examined by a Hitachi 7700 TEM (Hitachi, Tokyo, Japan) at 80 kV acceleration voltage and the images processed using ImageJ v1.5.2 [33].

### 2.5. Phage Host Range

The host range spectrum of each of the isolated phages was assessed using bacterial strains available in our lab and those sourced from the biobank mentioned above (Table 2), following the protocol reported by Kutter [34]. The results were then recorded as either being positive, intermediate (turbid) or negative.

### 2.6. Assessment of Phage Stability at Different Parameters

The protocol of Verma et al. [35] was applied with minor modifications to determine the resilience of the isolated phages at different pH and temperature. Briefly, 1 M NaOH and 1 M HCl was used to adjust the pH of 6 tubes containing 9 mL nutrient medium to 2, 4, 6, 8, 10 and 12. Then 1 mL of Pseudomonas phage_AUS034 with a titre of 1.3 × 10^4^ pfu/mL was added into each of the tubes. After incubation for 2 h, 50 μL of the phage-containing solution was mixed with 100 μL of a suspension of mid exponential phase *P. aeruginosa* AUS34 and assayed in triplicate with the overlay agar method to determine the remaining phages in the solution, expressed as percentages. In assessing the phages’ stability at different temperatures, tubes containing the phage (1.3 × 10^4^ pfu/mL) were placed in a water bath at 95 °C, 85 °C, 75 °C, 65 °C, 55 °C and 45 °C for 60 min. Aliquots of 2 μL of each of these phages were then added to 100 μL of mid-exponential *P. aeruginosa* and assayed in triplicate by the overlay agar method as mentioned above. This protocol was applied for the rest of the isolated phages and their corresponding bacterial strains.

### 2.7. Adsorption Assay

An adsorption assay was performed using the protocol by Gallet et al. [36] with modifications. Briefly, approximately 3.3 × 10^6^ PFUs of Pseudomonas phage_AUS034 was added to a tube containing approximately 3.3 × 10^8^ CFUs of *P. aeruginosa* AUS34 in 10 mL of pre-warmed LB medium. This was then placed in an incubator at 37 °C and 100 rpm for 30 min. Then 250 mL of the culture was sampled at intervals of 5 min, filtered through a 0.22 μm membrane filter and plated using the plaque assay as above. This was done three times and the average remaining phages plotted as a percentage of the original concentration. The adsorption constant was calculated as described by Kropinski [37]. This protocol was applied for the rest of the isolated phages and their corresponding bacterial strains. The adsorption was determined at 10 min for Pseudomonas Phage_AUS260 and Pseudomonas Phage_AUS391 and at 20 min for Pseudomonas Phage_AUS301 and Pseudomonas Phage_AUS034.

### 2.8. One Step Growth

A one-step growth experiment to ascertain the latent period (the time from adsorption to lysis of the bacterial cell) and burst size (number of phage progeny released per infected bacterial cell) of the isolated phage was performed as outlined by Kropinski [38] with modifications. Briefly, *P. aeruginosa* AUS34 was grown overnight to obtain a mid-exponential bacterial culture. This was then mixed with the Pseudomonas phage_AUS034 at a multiplicity of infection (MOI) of 0.01 in an adsorption tube. This was vortexed briefly and incubated at 37 °C for 10 min for adsorption to occur. The tube was then centrifuged at 10,000× *g* for 10 min and resuspended in fresh prewarmed medium. This was then placed in an incubator at 37 °C. Samples were collected from this at intervals of 10 min for 140 min and assayed in triplicate by the double agar overlay method. The latent period and burst size were estimated from enumeration of the plates after incubation for 24 h at 37 °C. The burst size was calculated by dividing the average phage number in stationary phase (burst) by the number of infecting phages (=average of phage in the latent phase − average of phage in the adsorption tube).

This protocol was applied for the rest of the isolated phages, with the sampling duration being 200 min for Pseudomonas phage_AUS260, 220 min for Pseudomonas phage_AUS301 and 230 min for Pseudomonas phage_AUS391.

### 2.9. Phage Cocktail Killing Action

The killing action of phage cocktails consisting of a combination of two, three or four phages was evaluated against 4 of the clinical strains used to isolate the phages as outlined by Chen et al. [39] with modifications. Briefly, 11 phage cocktails were designed (Table 3).

The amount of 5 × 10^6^ cfu of the phages or of the cocktail was mixed with a mid-exponential culture of each of the bacteria at an MOI of 0.01 in a flat-bottomed 96-well plate. The killing action against each of the bacterial strains was assessed independently in triplicate for 24 h in a CLARIOstar plate reader (BMG Labtech, Ortenberg, Germany) at OD600. Each assay had a bacterial control with the bacteria alone, a phage control with the phages and the original bacteria used in its isolation (ControlBPo, Clark, Philippines), a blank consisting of the LB broth medium only and finally the phage cocktails against the bacteria. Readings were taken every 5 min for 24 h. The phage killing curves were plotted hourly over 24 h.

### 2.10. Phage Genome Extraction, Sequencing and Analysis

#### 2.10.1. Phage DNA Extraction and Nuclease Treatment

Phage DNA was extracted using the Favorprep (Favorgen, Taiwan, China) viral nucleic acid extraction kit according to the manufacturer’s instructions (available on http://www.favorgen.com/favorgen/serv_1/mem_t1/h_1/pdf/rna/FAVNK%20000-1%20001%20001_1%20001-2_1503.pdf accessed on 23 October 2020) A Qubit spectrophotometer (Invitrogen, Waltham, MA, USA) and a NanoDrop 2000 system (Thermofisher Scientific, Waltham, MA USA) were used to assess the quality and quantity of the extracted nucleic acid. To ascertain the type of nucleic acid in the phage genomes, DNase I and RNase I (Invitrogen, Waltham, MA, USA) digestions were performed using the manufacturer’s protocol and visualised on a 0.8% agarose gel (Sigma Aldrich, Burlington, MA, USA) stained with a SYBR Safe in-gel stain (Invitrogen, Waltham, MA, USA) and ran at 45 V for 1 h.

#### 2.10.2. Library Preparation and DNA Sequencing

For the sequencing procedure, 50 μL of the extracted DNA from each of the phages was submitted in triplicate to the Australian Centre for Ecogenomics (ACE). Here, paired-end libraries were prepared from the extracted DNA samples using the Nextera XT DNA Library Preparation Kit (Illumina, San Diego, CA, USA) according to the manufacturer’s protocol. The DNA libraries were then sequenced on the NextSeq 500 (Illumina, San Diego, CA, USA) platform to generate 1 GB of 2 × 150 bp paired end reads per sample.

#### 2.10.3. Bioinformatic Analysis of the Phage Genomes

Quality control of the sequenced reads was done using FastQC v0.11.9 [40] and Trimmomatic v0.39 [41] with the following parameters: HEADCROP:20, MINLEN:100 AVGQUAL:30 and the rest left as default. Only overlapping read pairs were kept. Raw sequencing reads were assembled separately for each sample using SPAdes v3.12.0 in metagenomic mode with k-mer size settings of -k 29, 59 and 67 [42]. VirSorter v1.0.3 was used to identify putative virion contigs (VirSorter categories 1, 2 and 3) using both database options: -db 1 (RefSeq viruses) and -db 2 (viromes) [43]. FGENE SV0 [44], GeneMarkS v4.28 [45], RASTtk v1.3.0 [46] and Prodigal v2.6.2 [47] were applied for prediction of coding sequences (CDSs) with the genes predicted with at least three of them being considered for further analysis. Functional annotation on the detected domains was performed using Blast2Go [48] and blastn [49]. In the Geneious Prime v. 2020.2 [50] (http://www.geneious.com/ accessed on 19 December 2020) suite, a tRNA gene search was performed using tRNAscan-SE v. 2.0 [51]. Finally, a tandem repeat search by Tandem Repeats finder [52] was done. The results from these analyses were all curated manually.

#### 2.10.4. Phylogenetic Analysis

The whole genome nucleotide sequences of the phages and their five most similar sequences as per the NCBI nucleotide database (nr/nt) [49] were used to construct a phylogenetic tree with VICTOR (https://victor.dsmz.de accessed on 22 December 2020) using the Genome-BLAST Distance Phylogeny (GBDP) method and bootstrap support values from 100 replications [53]. This was then visualised and edited by FigTree v1.4.4 [54] and Treedyn v198.3 [55].

#### 2.10.5. Comparative Genomics

Artemis v18.1.0 [56], Easyfig v2.2.2 [57] and the BLASTn algorithm [49] were used for genomic synteny comparisons and visualisation of genome alignments of the four isolated bacteriophages.

#### 2.10.6. Nucleotide Sequence Accession Numbers

The nucleotide sequences of the bacteriophages’ genomes were deposited in GenBank under the accession numbers MW512831, MW512832, MW512833 and MW512834 for Pseudomonas Phage_AUS034, Pseudomonas Phage_AUS260, Pseudomonas Phage_AUS301 and Pseudomonas Phage_AUS391, respectively.

## 3. Results

### 3.1. Isolation and Morphology of the Phages

From the enriched activated sludge sample, four bacteriophages were isolated against *P. aeruginosa* AUS34, *P. aeruginosa* AUS260, *P. aeruginosa* AUS301 and *P. aeruginosa* AUS391, referred to, respectively, as Pseudomonas Phage_AUS034, Pseudomonas Phage_AUS260, Pseudomonas Phage_AUS301 and Pseudomonas Phage_AUS391 (Table 4 and Appendix A).

The TEM micrographs (Figure 1) and the morphologies observed below are in line with Pseudomonas Phage_AUS034, being a putative member of the family Podoviridae, Pseudomonas Phage_AUS260 of the family Siphoviridae and the last two Pseudomonas Phage_AUS301 and Pseudomonas Phage_AUS391 of the family Myoviridae.

### 3.2. Stability at Different Temperatures and pH

Pseudomonas Phage_AUS391 was the most unstable at higher temperatures, being inactivated beyond 55 °C, while the rest were slightly tolerant beyond 70 °C (Figure 2). All the phages were stable at neutral and alkaline pH but very unstable below pH of 6 (Figure 3).

### 3.3. One-Step Growth Dynamics of the Isolated Bacteriophages

Based on the one-step growth curves of the bacteriophages (Figure 4), the latent period in minutes and the burst sizes in PFU/cell were determined (Table 5).

### 3.4. Host Range Analysis

None of the phages isolated was able to infect another strain of *Pseudomonas* other than those of *P. aeruginosa* (Table 6). Pseudomonas Phage_AUS034 and Pseudomonas Phage_AUS391 were able to cause lysis in five strains of *P. aeruginosa*, Pseudomonas Phage_AUS260 two strains of *P. aeruginosa* and Pseudomonas Phage_AUS301 four strains of the isolates challenged.

### 3.5. Adsorption Assay

The adsorption characteristics of the isolated phages and their adsorption constant k (mL/min) at the first 5 min are presented in Table 7 and Figure 5, respectively.

### 3.6. Phage Cocktail Killing Action

The killing action of the cocktails varied quite significantly between the strains of *P. aeruginosa* for the 24-h duration of the experiment (Figure 6). Pseudomonas Phage_AUS34 lysed a higher number of the host cells than any of the cocktails for the 24 h. For the case of *P. aeruginosa* AUS260 and *P. aeruginosa* AUS391, only cocktail 5 underperformed with the rest of the cocktails performing better than Pseudomonas Phage_AUS260 and Pseudomonas Phage_AUS391. As for the case of *P. aeruginosa* AUS301, apart from cocktail 5, no clear killing action advantage was observed in using the cocktails, as they performed almost similarly to Pseudomonas Phage_AUS301.

### 3.7. Analysis of Bacteriophage Genomes

The sequenced genomes of the four phages were characterised (Table 8). Each of them had a dsDNA genome based on the nuclease digestion (Appendix A). The approximate sizes of their genomes ranged between 42 kb and 65 kb. The GC content for all the phages ranged between 52.2 and 58.8% (Table 8). In total, 367 opening reading frames (ORFs) were predicted between the four genomes, with 269 of these being coding sequences (CDS); 109 of the CDS were assigned putative functions (Appendix A). These included DNA replication (e.g., DNA polymerase III epsilon subunit, DNA polymerase III alpha subunit, DNA polymerase II, DNA ligase and DNA helicase), nucleotide metabolic processes (e.g., glutamine-fructose-6-phosphate aminotransferase, putative gamma-glutamyl cyclotransferase and putative acetyl transferase), DNA packaging (terminase small subunit and terminase large subunit), structural proteins involved in phage head assembly and tail assembly (e.g., head-to-tail connector complex protein, head morphogenesis protein, tail fibre proteins, capsid and scaffold protein, baseplate plate component, tail fibre proteins) and lytic processes (e.g.,T4-like lysozyme, SAR endolysin glycosyl hydrolase, putative lytic tail protein, putative endolysin and exonuclease). Homology searches of the ORFs did not reveal any lysogenic genes or host genome-related sequences; hence, we inferred that they were all lytic phages. Genes encoding toxins, host virulence factors or those associated with antibiotic resistance in *P. aeruginosa* were also absent.

Genome comparisons between the phage genomes revealed that Pseudomonas Phage_AUS034 (Podoviridae) and Pseudomonas Phage_AUS260 (Siphoviridae) were neither related to one another nor to the rest of the phages. The remaining two, Pseudomonas Phage_AUS301 and Pseudomonas Phage_AUS391, had a high degree of similarity (Figure 7), both belonging to the Myoviridae.

Table 9 summarises the highest GenBank database matches for each of the isolated phages based on whole genome alignment. The genome sequence of Pseudomonas Phage_AUS034 was most similar to Pseudomonas phage oldone, an unclassified Bruynoghevirus which is also of the family Podoviridae. Pseudomonas Phage_AUS260 had the highest similarity with another member of the family Siphoviridae, a Septimatrevirus Xanthomonas phage Samson. As for the two myoviruses isolated in this study, Pseudomonas Phage_AUS301 and Pseudomonas Phage_AUS391, they were most similar to Pseudomonas phage S50 and Pseudomonas virus Pa193, respectively, both reported as unclassified Pbunaviruses (Table 9).

A phylogenetic analysis (Figure 8) showed that Pseudomonas Phage_AUS301 and Pseudomonas Phage_AUS391 clustered together with other phages of the family Myoviridae, specifically with other Pbunaviruses. Pseudomonas Phage_AUS034 was closely related to members of the family Podoviridae and genus *Bruynoghevirus*, and Pseudomonas Phage_AUS260 to those belonging to the family Siphoviridae and genus *Septimatrevirus* (with both genera currently unclassified).

## 4. Discussion

With no promising antibiotics available in the pipeline for the WHO critical priority MDR pathogens, including *P. aeruginosa* [6], alternative strategies are required urgently to mitigate the burden in cost and suffering due to chronic infections caused by this opportunistic pathogen [58,59]. According to a number of reviews [60,61,62,63], phage therapy is one such alternative warranting investigation and development. In this study, we isolated four phages against clinical stains of *P. aeruginosa* isolated from the sputum of CF patients, from blood and joint exudate. Activated sludge was used here as an abundant and broadly diverse source for isolation of novel phages, which were subsequently characterised towards assessing their suitability as potential candidates for phage therapy. Three of the phages had a broad host range, being able to cause lysis in numerous strains of *P. aeruginosa* they were challenged against, where Pseudomonas Phage_AUS034 and Pseudomonas Phage_AUS391 caused lysis in five of the eleven strains tested, and Pseudomonas Phage_AUS301 was capable of lysing four of them.

All the phages showed optimum growth at 37 °C, which is also the optimum growth temperature of their clinical *P. aeruginosa* host strains [29] and is the average normal human body temperature [64]. This is very relevant if phage therapy is to be conducted in vivo. The pH of the activated sludge source used for phage isolation was 7.4, close to the observed optimum pH 8 for phage production, and the maximum titre of the isolated phages was indeed observed at a pH of 8. According to Jayaraman et al. [65], the pH for submucosal gland secretions in normal airways and in CF patients do not greatly vary and range between 6.5 and 7.6, while the pH for blood and synovial fluid is about 7.4 [66,67]. Thus, there would be no significant issues as far as pH is concerned for the use of these phages in clinical applications. The burst sizes of 143, 121, 86 and 51 of the isolated phages are relatively large and present an advantage in phage therapy, with a higher likelihood of the released phages able to reach their infection sites [68], thus increasing their efficacy. However, Hyman stated in a review [69] that phages with a long latent period were less desirable as agents for phage therapy. This observation is relevant to these results since the isolated phages had relatively long latent periods of 50, 60, 80 and 90 min; thus, optimisation of the conditions may be critical to improve the chances of success.

In 2012, Anderson [70] found that the use of phage cocktails was advantageous against *P. aeruginosa* in the lungs of CF patients. This was later corroborated by Hraiech et al. [71] and is likely because cocktails act by increasing the host range of the treatment [72]. Moreover, cocktails have been reported to help reduce bacterial resistance in cases where bacterial biofilms form [73]. In line with this, the use of cocktails in our study was advantageous in three of the four bacteria studied. Only in the case of *P. aeruginosa* AUS034 did the single phage originally isolated using this strain perform better than all the cocktails designed. The reason for this phenomenon remains unclear to us and may warrant further investigation.

The genomes of the isolated phages were relatively small (approximately 42 kb to 65 kb) in tandem with their capsid size as reported by Dion et al. [74] to be the case in most studied Caudovirales. Synteny comparative genomics on the whole genome sequences of these phages confirmed the families and genera identified through morphology observation above, with a high degree of similarity between Pseudomonas Phage_AUS301 and Pseudomonas Phage_AUS391 (both being myoviruses within the genus *Pbunaviridae*). These two phages revealed no relationship with Pseudomonas Phage_AUS034 (a member of the Podoviridae family and the genus *Bruynoghevirus*) and Pseudomonas Phage_AUS260 (a siphovirus from the genus *Septimatrevirus*). The results of the host range analysis of the novel phage isolates revealed some of the phage isolates can infect other hosts aside from their original strain, e.g., the podovirus Pseudomonas Phage_AUS301 can also infect the original host of the siphovirus Pseudomonas Phage_AUS260. This indicates these different phages may identify the same bacterial surface receptors for phage tail fibres and adsorption, the first step of phage infection. As reported, revealing key target host receptors in phage infection is key to future design of phage cocktails and therapeutic treatments [75]. Further investigation of the role of target receptors and the potential mechanisms leading to phage resistance is therefore warranted for future studies.

Genome annotation of the novel isolated bacteriophages showed that they had all basic structural and functional phage genetic elements encoding host lysis, DNA replication and modification, nucleotide metabolic processes, phage head assembly proteins and tail structural assembly proteins among other functional proteins. A further analysis of their genomes was performed using PhageLeads (http://130.226.24.116/phagecompass/index.html#/PhageLeads accessed on 23 January 2020) to ascertain the absence of undesirable genes (as potential candidates for use in phage therapy) such as lysogenic genes and host genome-related sequences encoding toxins, host virulence factors and antibiotic resistance among others [67,76,77,78,79].

Pseudomonas Phage_AUS034 and Pseudomonas Phage_AUS260 had tRNA sequences in their genomes. The presence of tRNAs is frequently identified in Myoviridae-like phages with large genomes [80]. The role of tRNA proteins translated from phage genes, based on phage codon bias that differs from the bacterial host codon usage, enables the synthesis of unique phage viral proteins [81,82]. The two phage genomes identified in our study that contain tRNA genes highlight the novel role of tRNAs in phage genomes, directing protein translation towards phage-related products, not that of their hosts.

Bacteriophages can be engineered to include genes coding for antibacterial compounds targeting MDR bacteria [83,84]. Such techniques are used to modify or increase the antibacterial capabilities of bacteriophages. Most bacteriophages have high host specificity and thus a narrow host range [69]. This is a limitation we encountered with the isolated phages. Whereas this is good, in that beneficial normal flora of the body is not affected, infections by multiple strains or species of bacteria can be very challenging for phages with a narrow host range [69]. To alleviate this, genes coding for receptor-binding proteins can be targeted by genetic engineering to expand the host range such as by adding a heterologous receptor binding domain [83]. The CRISPR-Cas system in bacteria is a rudimentary form of immune system that enables bacteria to cleave phage DNA that may render phage therapy useless [77]. Modifications to nucleotides mediating CRISPR-Cas complex binding/cleavage [85] or addition of genes coding for the expression of anti-CRISPR proteins [86] have been explored in phages against the CRISPR-Cas system. Other applications have been reviewed elsewhere by Gibb et al. [84] that include the use of phage-based genetic engineering techniques in altering antibiotic sensitivity in susceptible bacteria, disrupting biofilms by using phage-encoded depolymerases and killing bacteria using phage-encoded endolysins. These methods, though beyond the scope of this study, are relevant and thus should be considered in future studies.

The four novel phages described in this study represent important additions to a repository of characterized phage candidates, particularly for treatment of *P. aeruginosa* strains that occur in Australian clinical settings. Furthermore, these new phage isolates will be now available for addition to a national phage biobank proposed for Australia and beyond its borders. Adams [87], and later Hyman [69], noted that though phage therapy is promising, there may be differences when such is to be applied against biofilms or in patients, but a controlled lab environment is a safe start before complex systems can be explored. A review by Rossitto et al. [20] outlines many of the challenges associated with development of phage therapy, some of which this study sought to address. Further studies have to be conducted to obtain more information about the features and performance of the isolated phages towards practical application in phage therapy.

## 5. Conclusions

In this study we isolated a number of phages against MDR clinical strains of *P. aeruginosa* isolated from Australian clinical environments and determined these phages had qualities that made them amenable for use as candidates for phage therapy against the targeted clinical isolates. Phages isolated against such clinical isolates are few and there are none so far isolated that are specific to the Australian clinical isolates of *P. aeruginosa* used in this study. Though there are many challenges against phage therapy, isolation and characterisation of novel phages is a crucial step in the advancement of phage therapy. Based on the findings of this study, the isolated phages seem promising since they display desirable qualities considered indispensable among candidates for phage therapy, such as adequate temperature and pH optima, high burst size and the absence of undesirable genes (lysogenic genes and host genome-related sequences encoding toxins, host virulence factors and antibiotic resistance). The discovery of novel phages is expected to aid in the development of alternative but efficient intervention strategies against chronic life-threatening strains of *P. aeruginosa*. With no promising new antibiotic currently in the pipeline against these clinical strains, phage therapy using the novel bacteriophages presented herein represents a viable alternative. Further research in animal models or other systems that mirror the involved human organ systems are required to fully assess their potential in vivo.

## Figures and Tables

**Figure 1 microorganisms-10-00210-f001:**
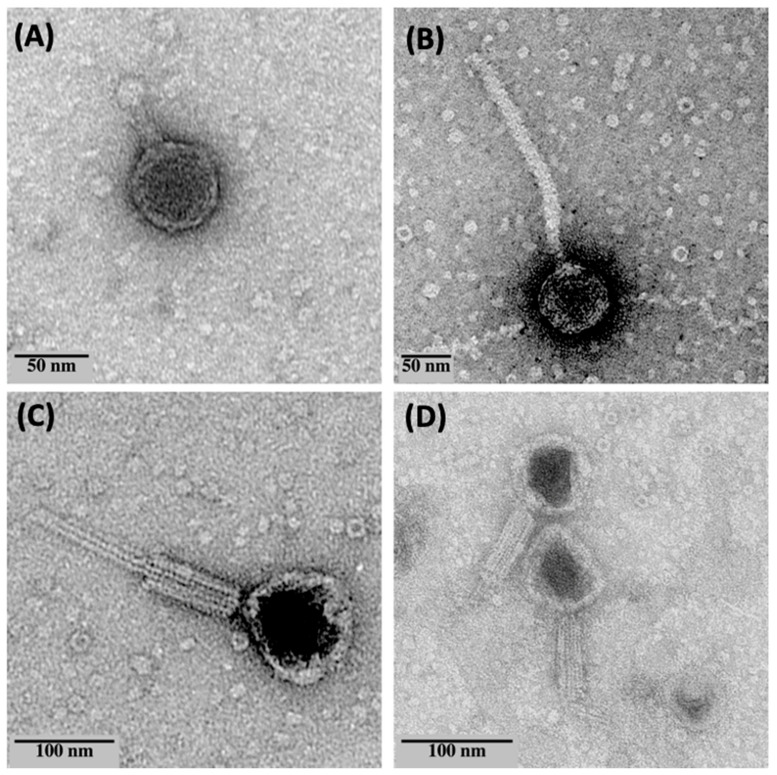
TEM micrographs of (**A**) Pseudomonas Phage_AUS034, (**B**) Pseudomonas Phage_AUS260, (**C**) Pseudomonas Phage_AUS301 and (**D**) Pseudomonas Phage_AUS391.

**Figure 2 microorganisms-10-00210-f002:**
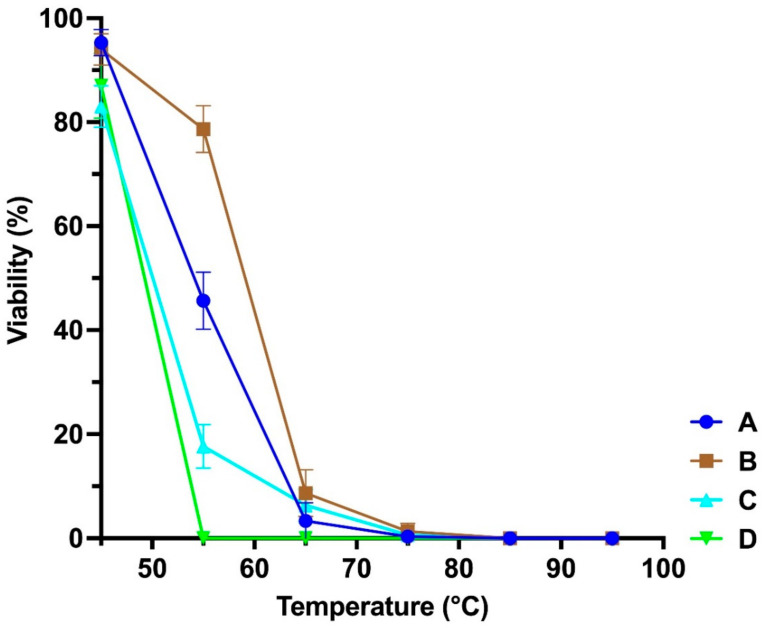
Thermal stability of bacteriophages: (A) Pseudomonas Phage_AUS260, (B) Pseudomonas Phage_AUS391, (C) Pseudomonas Phage_AUS301 and (D) Pseudomonas Phage_AUS034.

**Figure 3 microorganisms-10-00210-f003:**
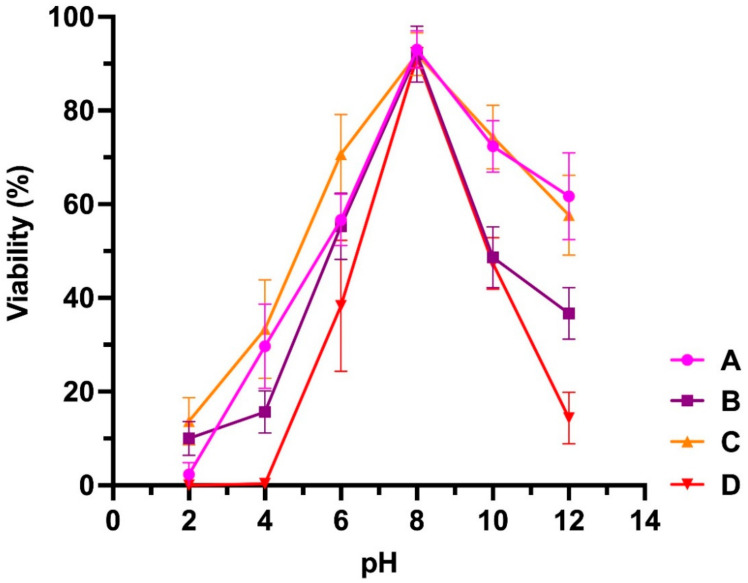
Sensitivity of bacteriophages to pH. (A) Pseudomonas Phage_AUS034, (B) Pseudomonas Phage_AUS301, (C) Pseudomonas Phage_AUS260 and (D) Pseudomonas Phage_AUS391.

**Figure 4 microorganisms-10-00210-f004:**
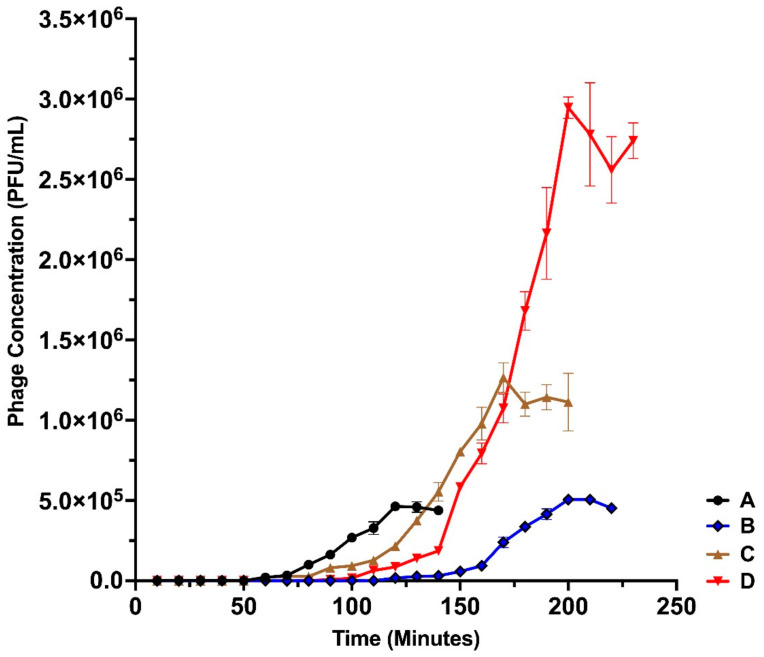
One-step growth curve of bacteriophages: (A) Pseudomonas Phage_AUS034, (B) Pseudomonas Phage_AUS260, (C) Pseudomonas Phage_AUS301 and (D) Pseudomonas Phage_AUS391.

**Figure 5 microorganisms-10-00210-f005:**
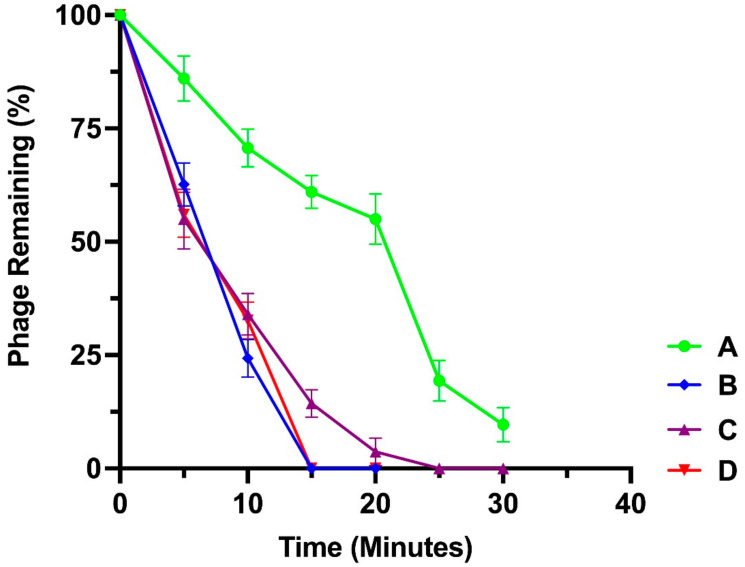
Adsorption characteristics of bacteriophages: (A) Pseudomonas Phage_AUS34, (B) Pseudomonas Phage_AUS301, (C) Pseudomonas Phage_AUS260 and (D) Pseudomonas Phage_AUS391.

**Figure 6 microorganisms-10-00210-f006:**
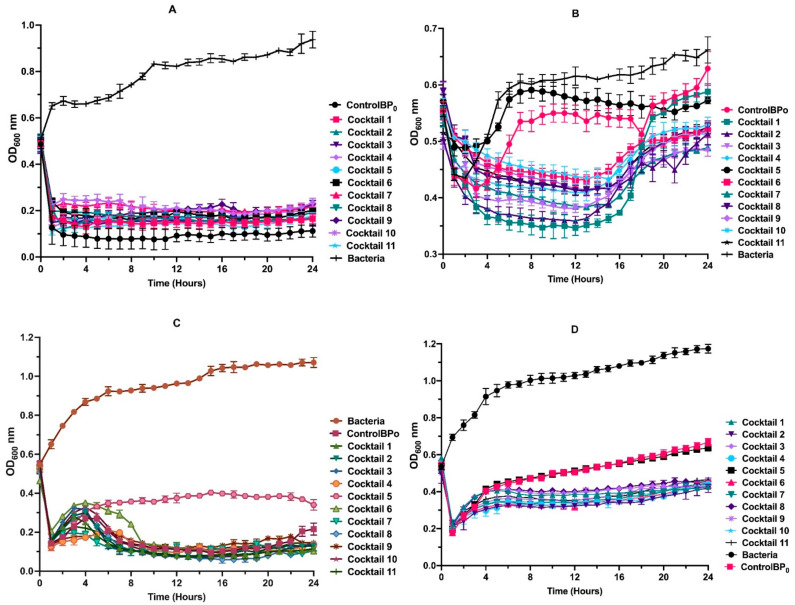
Phage cocktail killing action against (**A**) *P. aeruginosa* AUS34, (**B**) *P. aeruginosa* AUS260, (**C**) *P. aeruginosa* AUS301 and (**D**) *P. aeruginosa* AUS391 for 24 h. Bacteria (negative control) consists of bacteria without phage treatment, Control BPO (positive control) bacterial treatment with native phage. OD: optical density.

**Figure 7 microorganisms-10-00210-f007:**
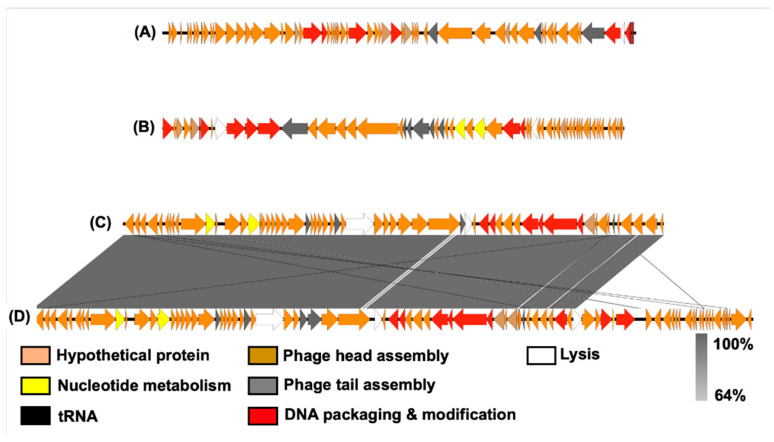
Diagram showing the alignment and genomic synteny comparison between the four phages: (**A**) Pseudomonas Phage_AUS034, (**B**) Pseudomonas Phage_AUS260, (**C**) Pseudomonas Phage_AUS301 and (**D**) Pseudomonas Phage_AUS391. The coding sequences identified are represented by the arrows.

**Figure 8 microorganisms-10-00210-f008:**
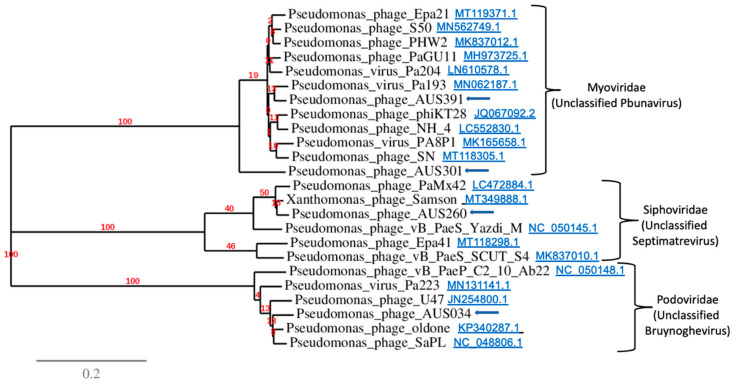
A phylogenetic tree constructed using whole genome sequences of the isolated phages together with the five most homologous sequences to each of these four, obtained from the GenBank database, using the Genome-BLAST Distance Phylogeny (GBDP) method and bootstrap support values from 100 replications.

**Table 1 microorganisms-10-00210-t001:** Clinical strains of *P. aeruginosa* used to isolate new phages in this study.

Clinical Strain	Isolation Source
*P. aeruginosa* AUS34	Sputum
*P. aeruginosa* AUS260	Sputum
*P. aeruginosa* AUS301	Blood
*P. aeruginosa* AUS391	Joint aspirate

**Table 2 microorganisms-10-00210-t002:** Bacterial strains used for host range determination.

Bacteria	ID (PubMLST) or Accession Number (ACCN)	Source
*P. aeruginosa* AUS432	ID: 1249	Biobank at CFS-QCH Brisbane, Australia
*P. aeruginosa* AUS247	ID: 1065	Biobank at CFS-QCH Brisbane, Australia
*P. aeruginosa* AUS391	ID: 1209	Biobank at CFS-QCH Brisbane, Australia
*P. aeruginosa* AUS34	ID: 580	Biobank at CFS-QCH Brisbane, Australia
*P. aeruginosa* AUS291	ID: 1109	Biobank at CFS-QCH Brisbane, Australia
*P. aeruginosa* AUS23	ID: 569	Biobank at CFS-QCH Brisbane, Australia
*P. aeruginosa* AUS455	ID: 1272	Biobank at CFS-QCH Brisbane, Australia
*P. aeruginosa* AUS260	ID: 1078	Biobank at CFS-QCH Brisbane, Australia
*P. aeruginosa* AUS301	ID: 1119	Biobank at CFS-QCH Brisbane, Australia
*P. aeruginosa* AUS403	ID: 1220	Biobank at CFS-QCH Brisbane, Australia
*P. aeruginosa* PAO1	ACCN: NC 002516	Lab strain (ACWEB)
*Pseudomonas putida* KT2440 (RP4 plasmid)	ACCN: AE015451	Lab strain (ACWEB)

**Table 3 microorganisms-10-00210-t003:** Phage cocktails used against the strains of *P. aeruginosa*.

Cocktail	Phage Constituents of the Cocktails
Cocktail 1	Pseudomonas Phage_AUS034 + Pseudomonas Phage_AUS260
Cocktail 2	Pseudomonas Phage_AUS034 + Pseudomonas Phage_AUS301
Cocktail 3	Pseudomonas Phage_AUS034 + Pseudomonas Phage_AUS391
Cocktail 4	Pseudomonas Phage_AUS260 + Pseudomonas Phage_AUS301
Cocktail 5	Pseudomonas Phage_AUS260 + Pseudomonas Phage_AUS391
Cocktail 6	Pseudomonas Phage_AUS301 + Pseudomonas Phage_AUS391
Cocktail 7	Pseudomonas Phage_AUS034 + Pseudomonas Phage_AUS260 + Pseudomonas Phage_AUS301
Cocktail 8	Pseudomonas Phage_AUS034 + Pseudomonas Phage_AUS260 + Pseudomonas Phage_AUS391
Cocktail 9	Pseudomonas Phage_AUS260 + Pseudomonas Phage_AUS301 + Pseudomonas Phage_AUS391
Cocktail 10	Pseudomonas Phage_AUS034 + Pseudomonas Phage_AUS301 + Pseudomonas Phage_AUS391
Cocktail 11	Pseudomonas Phage_AUS034 + Pseudomonas Phage_AUS260 + Pseudomonas Phage_AUS301 + Pseudomonas Phage_AUS391

**Table 4 microorganisms-10-00210-t004:** Phages isolated using the clinical bacterial strains.

Isolated Phage	Bacteria	Plaque Diameter (mm)	Phage Size (nm)	
			Head	Tail
Pseudomonas Phage_AUS034	*P. aeruginosa* AUS34	3	56 ± 0.5	45 ± 2
Pseudomonas Phage_AUS260	*P. aeruginosa* AUS260	1.5	58 ± 0.5	190 ± 1
Pseudomonas Phage_AUS301	*P. aeruginosa* AUS301	0.6	73 ± 1	70 ± 2 (Contracted) 132 ± 2 (Non-contracted)
Pseudomonas Phage_AUS391	*P. aeruginosa* AUS391	0.6	75 ± 2	81 ± 2 (Contracted) 121 ± 3 (Non-contracted)

**Table 5 microorganisms-10-00210-t005:** The latent period and burst size of the isolated phages.

Isolated Phage	Latent Period (Minutes)	Burst Size (PFU/Cell)
Pseudomonas Phage_AUS034	50	143 ± 16
Pseudomonas Phage_AUS260	60	86 ± 7
Pseudomonas Phage_AUS301	80	121 ± 11
Pseudomonas Phage_AUS391	90	51 ± 4

**Table 6 microorganisms-10-00210-t006:** Host range analysis of the isolated bacteriophages.

Bacteria	Lytic Activity of Phage			
	Pseudomonas Phage_AUS034	Pseudomonas Phage_AUS260	Pseudomonas Phage_AUS301	Pseudomonas Phage_AUS391
*P. aeruginosa* AUS432	+ +	− −	+ −	− −
*P. aeruginosa* AUS247	− −	− −	− −	− −
*P. aeruginosa* AUS391	− −	+ +	− −	+ +
*P. aeruginosa* AUS034	+ +	− −	+ +	+ +
*P. aeruginosa* AUS291	− −	+ −	+ −	+ +
*P. aeruginosa* AUS023	− −	− −	− −	− −
*P. aeruginosa* AUS455	+ −	+ −	+ −	+ +
*P. aeruginosa* AUS260	+ +	+ +	+ +	+ −
*P. aeruginosa* AUS301	+ +	+ −	+ +	+ −
*P. aeruginosa* AUS403	− −	− −	− −	− −
*P. aeruginosa* PAO1	+ +	− −	+ +	+ +
*Pseudomonas putida* KT2440 (RP4 plasmid)	− −	− −	− −	− −

Legend: (+ +) lysis, (+ −) incomplete/intermediate lysis (turbid), and (− −) no lysis.

**Table 7 microorganisms-10-00210-t007:** The adsorption percentages and adsorption constants of the phage isolates.

Phage	Adsorption (%)	k (mL/min)
Pseudomonas Phage_AUS034	81	1.26 × 10^−8^
Pseudomonas Phage_AUS260	100	3.36 × 10^−9^
Pseudomonas Phage_AUS301	97	4.76 × 10^−9^
Pseudomonas Phage_AUS391	100	4.33 × 10^−9^

**Table 8 microorganisms-10-00210-t008:** Characteristics of the genomes of the isolated phages.

Bacteriophage Name	Genome Size (bp)	GC Content (%)	Number of ORFs	Number of CDSs	CDSs with Putative Functions	Number of tRNAs
Pseudomonas Phage_AUS034	44,167	52.2	75	64	19	4
Pseudomonas Phage_AUS260	42,020	54.7	78	54	28	1
Pseudomonas Phage_AUS301	49,455	55.8	98	62	27	0
Pseudomonas Phage_AUS391	65,515	55.5	125	89	29	0

**Table 9 microorganisms-10-00210-t009:** Phages with sequences most similar to the isolated phages in the GenBank database.

Isolated Phage	Most Similar to	Identity (%)	Query Cover (%)
Pseudomonas Phage_AUS034	Pseudomonas phage oldone	97.10	98
Pseudomonas Phage_AUS260	Xanthomonas phage Samson	95.45	99
Pseudomonas Phage_AUS301	Pseudomonas phage S50	98.75	100
Pseudomonas Phage_AUS391	Pseudomonas virus Pa193	97.59	98

## Data Availability

The data presented in this study are available within the article, while nucleotide sequences of the bacteriophages’ genomes were deposited in GenBank under the accession numbers listed above.

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
