# Peer review of "Novel Bacteriophages Show Activity against Selected Australian Clinical Strains of Pseudomonas aeruginosa"

_microorganisms, 2022, doi:10.3390/microorganisms10020210_

Round 1

Reviewer 1 Report

The manuscript by S. Namonyo et al is an interesting, clear and well-written paper; it describes the isolation and investigation of properties of several Pseudomonas phages that potentially could be used to combat pathogen-caused infections including infections during cystic fibrosis. This topic is of vital importance as frequently the therapeutic schemes used for the treatment of CF are not optimized, include significant doses of antibiotics, but are lack of effectiveness. It is evident that the approaches need to be modified significantly. Use of bacteriophages as independent pharmaceuticals or their combination with antibiotics could reduce the development of complications in infectious diseases caused by sensitive bacterial species.

On the other hand I am interested in the authors opinion about genetically engineered phages. Do authors suppose search for new native phages to be more perspective comparing to the modification and construction of artificial viruses with specified properties and without undesirable genes against different pathogens? Your ideas on this topic could be an additional paragraph in the Discussion part of the manuscript.

There are also several typos throughout the text: Lines 98, 101, 257 and 452 – Pseudomonas aeruginosa should be written in Italics. Line 152. Please, correct 1.3 x 104 pfu/mL.

I suppose, that this paper will be a solid contribution to the Microorganisms and will certainly appeal to its readers. The conclusions were verified by convincing data. It is recommended that this manuscript to be published in Microorganisms after completing minor revision

Reviewer 2 Report

# Burst size calculation formula (line 287) and figure 4, 6, and 7 looked blurry. hard to interpret the outcomes. 

The overall  science of this article is nicely organized, well hypothesized, complete story with logical flow.

Reviewer 3 Report

The manuscript is clear and interesting

Reviewer 4 Report

The authors isolate and characterize several phages infecting Pseudomonas bacteria, with the aim of providing candidate viruses for phage therapy in cystic fibrosis. The study is well written and conducted, and I have only a few points that should be resolved before publication.

  • All figures are pixelated, some beyond recognition (specifically Figs 4, 6 and 7) and need to be provided in higher resolution for proper review and publication.
  • In order to be suitable for phage therapy, the absence of antibiotic resistance genes and virulence factors should be investigated more thoroughly. I suggest the authors quickly run their genomes through appropriate databases (see Shen, A. and Millard, A., 2021. Phage Genome Annotation: Where to Begin and End. PHAGE, 2(4), pp.183-193 for a list)

Some additional line-by-line comments:

67: it looks like a rogue character after the word chemicals

69-73: This information is hopelessly outdated and needs to be updated for 2022. Many more genomes should be available by now.

101/Table 1/Table 2: Have these strains been sequenced/are accession numbers available? It would be helpful if they were added. If yes, the host range as indicated in Table 6 could be shown on a phylogenetic tree of the bacterial strains to give a better understanding of how different the infected strains are. Not absolutely necessary for publication of course.

187: The burst size formula is also very pixelated

257: P. aeruginosa is not italiziced. This is one of many instances (also in references) so the authors should go through the manuscript and fix it before publication.

281/Table 5: Can the authors provide standard deviations for burst size/latent period? Also, it should be noted in the text that these numbers only apply to the bacterial strains on which the phages were originally isolated. Numbers for other strains that they cross-infect would be interesting, but this is not necessary.

 295: Same as comment above

302: Since I can absolutely not read Figure 6 I cannot comment on the paragraph.
